Feature-based enhanced boosting algorithm for depression detection

Rohei Muhammad Sadiq 1
http://orcid.org/0000-0003-3421-4501 Varathan Kasturi Dewi 1 kasturi@um.edu.my
Palaiahnakote Shivakumara 2
http://orcid.org/0000-0003-4380-5303 Anuar Nor Badrul 3
1 Department of Information Systems, Faculty of Computer Science and Information Technology, Universiti Malaya , Kuala Lumpur , Malaysia
2 School of Science, Engineering & Environment, University of Salford , Salford, Manchester , United Kingdom
3 Department of Computer Systems & Technology, Faculty of Computer Science & Information Technology, Universiti Malaya , Kuala Lumpur , Malaysia
Alatas Bilal
Electronic publication date: 2025 Jul 29
Publication date: 2025
Volume: 11
Electronic Location ID: e2981
Received 2024 Jul 30; Accepted 2025 Jun 2
Copyright: © 2025 Rohei et al.
Copyright year: 2025
Copyright holder: Rohei et al.
License: This is an open access article distributed under the terms of the Creative Commons Attribution License, which permits unrestricted use, distribution, reproduction and adaptation in any medium and for any purpose provided that it is properly attributed. For attribution, the original author(s), title, publication source (PeerJ Computer Science) and either DOI or URL of the article must be cited.
License URL: https://creativecommons.org/licenses/by/4.0/

Keywords: Feature-based enhanced boosting algorithm, Depression detection, Enhanced boosting algorithm, Feature engineering

Funding: The authors received no funding for this work.

==============================
Depression is a rapidly increasing mental disorder that can interfere with a person’s ability and negatively affect functions in various aspects of life. Fortunately, machine learning and deep learning techniques have demonstrated excellent results in the early detection of depression using social media data. Most recently, researchers have utilized boosting algorithms including pre-defined boosting algorithms or built their own boosting algorithm for the detection of depression. However, both types of boosting algorithms struggle with the analysis of complex feature sets, the enhancement of weak learners, and the handling of larger datasets. Thus, this study has developed a novel feature-based enhanced boosting algorithm (F-EBA). The proposed model covers two pipelines, the feature engineering pipeline which improves the quality of features by picking up the most relevant features while the classification pipeline uses an ensemble approach designed to boost/elevate the model’s performances. The experimental results highlighted that various parameter including WordVec and BERT embeddings, attention mechanisms, and feature elimination techniques, significantly contributed to the selection of the most relevant features. This approach resulted in generating an optimized feature set that augmented both the model’s accuracy and its interpretability. In addition, utilizing over 46 million records, the F-EBA model significantly enhanced the performance of weak learners through a weight maximization strategy, achieving an impressive accuracy rate of 95%. Moreover, the integration of an adversarial layer that employs defense mechanisms against synonymous text and sarcastic phrases within the datasets has further boosted the F-EBA model’s accuracy to approximately 97%, surpassing the results reported in prior studies. Moreover, the optimized feature sets derived from the F-EBA model make a substantial contribution to boosting the performance of baseline classifiers, marking a novel advancement in the field.

Introduction

Depression, a prevalent mental health condition, involves irregularity in individuals’ behavior (Stein et al., 2020), and poses a significant public health problem around the globe, approximately 280 million people worldwide have depression (WHO, 2023). It is characterized by persistent feelings of sadness, hopelessness, and loss of interest in activities. Depression, caused by a variety of factors including family stress, unemployment, abuse, cultural tensions, social stigma, traits, and jealousy, not only impairs cognitive, emotional, and relational functioning but also affects all aspects of physical health (Telles-Correia, Saraiva & Gonçalves, 2018; Horwitz, 2020; Singh, 2021). Early and accurate detection of depression is vital for effective treatment and management of disorders. Despite this, researchers have recognized the potential of social media data as valuable resources and utilizing various predicting techniques for predicting depression (Chancellor & De Choudhury, 2020; Zhu et al., 2023; Chiong, Budhi & Dhakal, 2021; Dheeraj & Ramakrishnudu, 2021; Kour & Gupta, 2022). In this regard, machine learning algorithms, in particular boosting algorithms, have shown great potential in detecting depression accurately and efficiently.

Boosting algorithms, such as AdaBoost (Freund & Schapire, 1997), gradient-boosted decision trees (GBDT) (Friedman, 2001), extreme gradient boosting (XGBoost; XGB) (Chen & Guestrin, 2016), light gradient-boosting machine (LGBM) (Ke et al., 2017), and CatBoost (Guillen, Aparicio & Esteve, 2023), have revolutionized the field of supervised ensemble machine learning. These algorithms iteratively train weak classifiers and combine their results to create strong classifiers, offering superior predictive performance compared to individual classifiers. The evolution of boosting algorithms has been driven by the need for efficient handling of various data types and complexities. For instance, the introduction of LightGBM (Guillen, Aparicio & Esteve, 2023), GBDT (Friedman, 2001), and XGBoost (Chen & Guestrin, 2016) addressed the challenge of processing large datasets through parallel processing and optimized tree structures. CatBoost (Guillen, Aparicio & Esteve, 2023), on the other hand, specifically targets the handling of categorical features, which are common in real-world datasets. Moreover, researchers have extended the capabilities of existing boosting algorithms or devised novel ones to address specific challenges in diverse domains. These efforts have led to the development of self-made boosting algorithms tailored to particular problem domains, such as depression detection or addressing issues of overfitting and unbalanced datasets.

However, both pre-defined and self-made boosting algorithms frequently encounter challenges in effectively selecting and managing complex features, enhancing weak learners to mitigate error rates, demonstrating generalizability, and performing cross-validation across the entire dataset, as well as the inability to adequately handle synonymous text or sarcastic posts within datasets. This study investigates whether a novel feature-optimised boosting algorithm (F-EBA) can significantly enhance depression detection performance by addressing feature complexity, model robustness, and nuanced language such as sarcasm and synonymy in social media data. These limitations highlight the need for a more adaptive and resilient classification approach, capable of handling complex linguistic patterns and evolving real-world data inputs in the context of mental health detection.

Hence, this study developed novel boosting algorithms called F-EBA rooted based on CBPT (cost-sensitive boosting pruning tree) and Adaboost to mitigate the mentioned shortcomings in two distinct ways, with the aim of designing and evaluating F-EBA as a new boosting framework. Feature optimisation was carried out via the F-EBA feature engineering pipeline, which enhance feature selection by utilising Word2Vec and BERT embeddings, in conjunction with a self-attention mechanism and recursive feature elimination (RFE). It assigns importance scores to features, removes low-weight ones, and generates a refined, high-quality feature set or an optimised feature set.

Ten (10) models of each BERT and BiLSTM were trained and utilized individually within an innovative boosting structure in classification pipelines, the approach involves re-sampling and assigning higher weights to misclassified features identified by the preceding classifier. This enables the subsequent classifier to prioritize these samples, progressively eliminating misclassifications, lowering error rates, and improving the overall performance of weak learners. Additionally, an adversarial layer has been integrated into the model to strengthen and withstand it accurately when a manipulated input is given designed to confuse it or handle synonymous text or sarcasm posts within the datasets. This addition is particularly pertinent in situations where the data exhibits frequent noise, and the model is continuously evolving and gaining strength.

This study builds upon existing methodologies in predictive modelling for depression detection by introducing the F-EBA model, marking a significant advancement in custom-build or self-made boosting algorithms. It provides a substantial solution to the key limitations of state-of-the-art (SOTA) approaches by optimizing feature sets that can effectively handle complex data patterns, reducing error rates through the enhancement of weak learner performances, ability to handle synonymous text and sarcastic posts with defense mechanisms, and accommodating larger datasets. By implementing a novel feature-engineering pipeline and an advanced boosting mechanism, as well as utilizing a robust defense mechanism the F-EBA model replicates significant improvements in classification accuracy by precise predictions and model robustness, particularly when handling large datasets.

Through this replication and enhancement, this study offers valuable insights into the existing literature, presenting a more effective model for depression detection that can be utilized in various real-world scenarios. The rest of this article is structured as follows: “Related Works” reviews related work on boosting algorithms for depression detection. “Methodology” details the design of the feature-based enhanced boosting algorithm. “Ethical Considerations” presents implementation details and experimental results, followed by a discussion in “F-EBA Model Structure and Implementation”. Finally, “Conclusion” concludes the article, outlining potential directions for future research.

Motivation and contribution of the study

Recently, detecting and monitoring depression remains a major challenge in healthcare. Government entities, surveillance systems, and mental health professionals are actively seeking effective ways to identify individuals at risk of depression at an early stage, as providing timely intervention and treatment is crucial for improving outcomes.

Social media platforms offer a unique opportunity in this regard, as people often express their thoughts, feelings, and emotions through text, images, and other modalities on these platforms. This has spurred growing research interest in leveraging social media data and developing computational methods to analyse this data for potential markers of mental health conditions like depression. However, recently with the development of machine learning, deep learning and AI have been used to exploit complex patterns and relationships that exist within unstructured social media data. Boosting algorithms, which combine multiple weak prediction models into a powerful ensemble classifier, have emerged as a promising approach for this task.

Various pre-boosting techniques, such as AdaBoost, XGBoost, including custom built boosting methods, have been explored for depression detection on social media. However, these algorithms often struggle with challenges like handling diverse data types, reducing misclassification, and avoiding overfitting in high-dimensional, large-scale datasets. In addition, it remains a key objective to attain clinically acceptable accuracy while managing the higher error rates associated with boosting weak learners. The primary motivation for this study is to develop a novel boosting algorithm framework that addresses these limitations, and aimed at improving computational approaches for social media data analytics and text classification tasks that centres around depression detection as a case study. Thus, the main contributions of this study are: (1) This research introduces a new feature-enhanced boosting algorithm called F-EBA, which is specifically designed to address the challenging aspects of depression detection on social media platforms

(2) F-EBA employs an adversarial layer to enhance resilience against synonymous and sarcastic social media posts, and uses hyperparameter tuning to refine feature sets for improved accuracy. This approach strengthens mental health analytics by closing the gaps in existing approaches to boosting as well as providing a strong solution for depression detection on social media.

(3) The proposed F-EBA approach contributes a versatile framework that can be applied to other text classification tasks, paving the way for future research in related domains.

Related works

There is a growing interest in research on mental health issues, notably depression, marked by common abnormalities in people’s way of thinking, feelings, and behaviours (Stein et al., 2020). Mental illness, caused by a variety of factors including family stress, unemployment, cultural tensions, and social stigma (Telles-Correia, Saraiva & Gonçalves, 2018; Horwitz, 2020; Singh, 2021) requires early treatment because 70% of untreated individuals with depression may witness of worsening symptoms and are at risk of suicide (Shen et al., 2017). Early detection of depression can avoid unwanted consequences, providing appropriate treatment (Abdullah & Choudhury, 2018). However, clinical procedures for depressed patients, involving interviews and surveys conducted by medical professionals, are ineffective for early risk detection if the patient does not disclose their depression (Marcus et al., 2012; Antony & Barlow, 2020). Fortunately, social media platforms such as Twitter, Facebook, Sina Weibo etc., provide people with opportunities to express their feelings, thoughts, and moods. They serve as valuable resources and data disclosure for researchers who are actively engaged in predicting depression through various predictive techniques (Chancellor & De Choudhury, 2020).

Boosting techniques/algorithms

In the context of predictive techniques, machine learning (ML) algorithms stand out as the most widely used predictive techniques for recognizing various mental illnesses, including depression (Abd Rahman et al., 2020; Arya & Mishra, 2021). These techniques are categorized into supervised, unsupervised, and semi-supervised learning methods, each with its own set of algorithms and models. Supervised machine learning techniques such as decision tree (DT), logistic regression (LR) and support vector machine (SVM) are used alongside unsupervised techniques, such as clustering and dimensionality reduction, to develop classification models from labelled data or even unlabelled data. Benton, Mitchell & Hovy (2017) discussed the advanced approaches such as supervised ensemble methods, transfer learning and multi-task learning, which combine several predictive models for improving performances. To this end, boosting algorithms or supervised ensemble methods, currently at the forefront of machine learning, utilized by numerous researchers (Nandanwar & Nallamolu, 2021; Reseena Mol & Veni, 2022; Mohammed et al., 2021; Ding et al., 2022), and have shown outstanding results for depression detection. Boosting (generally means increasing the performance of weak learners) is a popular sequential ensemble learning algorithm used to convert weak learners into strong learners aiming to increase the accuracy of the model (Breiman, 1996). Boosting algorithms are widely used for image recognition, motion tracking, and speech recognition. As previously mentioned, boosting algorithms come in two types, both of which are utilized for various scenarios, including depression, as discussed below:

Pre-defined boosting algorithms for depression detection

There are several pre-defined boosting algorithms exist, including Adaboost developed by Freund & Schapire (1997), GBDT by Friedman (2001), XGB by Chen & Guestrin (2016), CatBoost by Yandex (2017), and LightGBM by Ke et al. (2017). These algorithms were employed in conjunction with other machine learning techniques across diverse datasets by various researchers for the detection of depression. In the studies conducted by Chiong et al. (2021) and Skaik & Inkpen (2020), the gradient boosting decision trees (GBDT) algorithm was utilized for depression detection. In the first study, GBDT contributed to increasing the accuracy score from 90% to 98%. Conversely, the second study reported that GBDT performed optimally in training, achieving an F1-score of 96%. Both studies encountered challenges related to handling complex features.

Similarly, XGboost, AdaBoost, and gradient boost were utilised by Reseena Mol & Veni (2022) who tried to boost the accuracy of their stacked ensemble model for depression detection. They applied different embedding techniques such as Word2Vec, FastText, and Glove for feature extraction. In the comparative analysis across all algorithms, XGBoost achieved the highest accuracy at 95.6%. The researcher prefers more precise detection of depression from tweets by combining deep learning techniques with word embedding model. By utilizing AdaBoost, stochastic gradient, and XGBoost classifiers along with other machine learning techniques, Nandanwar & Nallamolu (2021) aimed to predict depression on a Twitter dataset with 13,514 tweets. The experiments revealed that AdaBoost and gradient boost exhibited a notable F-score rate of 93.09% in the detection of depression, outperforming other algorithms. Likewise, an ensemble model of the gradient boosting algorithm and supervised ML (support vector machine (SVM), logistic regression (LR), decision tree (DT)) was applied to Shen et al. (2017) dataset for predicting depression (Chiong, Budhi & Dhakal, 2021). The experimental results revealed a notable 10.4% improvement in detection across various categories of depression symptoms when GB collaborated with LR. The lack of complex feature set analysis and cross-validation across the entire dataset is a limitation of this study. Another ensemble algorithm of deep forest and XGBoost has been applied by Xia et al. (2022) on the Aliyun dataset of 1,428 samples with 21 features and two classes, depressed and non-depressed with numbers of 238 and 1,190 users respectively. For the experiment, a 10-fold cross-validation with a scanning window of size seven and layer three was implemented. The result showed that XGBoost performed with an accuracy of 81% respectively.

Additionally, the XGBoost algorithm, in conjunction with random forest (RF), SVM, and naive Bayes, was implemented by Malviya, Roy & Saritha (2021) and Mali et al. (2021) to classify depressed users in Reddit and Kaggle datasets, respectively. The experimental findings revealed that XGBoost attained the highest accuracy scores of 91% and 83.49%, respectively. Both studies encountered challenges related to complex feature sets. Afzoon, Rezvani & Khunjush (2021) analyzed two datasets from social media (Twitter 1.6 million tweets + Reddit) by applying different ML algorithms besides the XGBoost. Sentiment features along with two types of multimodal features (textual and temporal), and domain-related features were employed to train the model. As a result, XGBoost showed high performance and efficient results as compared to LR, NB and RF on both datasets with F-score of 77% and 87%. It was proposed that using deep multi-model domain-related features can improve efficiency. Semwal et al. (2023) detect depression by applying classical methods including gradient boosting (GB) and modality fusion on two datasets collected from Twitter which contained 6.1 and 4.6 million tweets, respectively. For feature extraction tokenization, WordNet, LIWC, and 2,304-unit vectors are used as inputs to train the model. The result showed that the GB algorithm had an accuracy of 52.38% with the word 3-gram. Adarsh et al. (2023) applied XGBoost with other ML algorithms to solve the unexpected bias in previous studies. The authors compared extreme gradient (XG), RF, SVM, DT and convolutional neural network (CNN), thus the result demonstrated that accuracies have been improved with noise correction and imbalance reduction by 5%. Finally, the accuracy of XG was 81.46% among the other ML algorithms and the authors suggested that it can be improved more by using multiclass classification.

In a recent study by Tai et al. (2024), the performance of SVM, RF, and LightGBM in categorizing sleep stages for depression patients and healthy volunteers was evaluated. LightGBM achieved the highest accuracy, with 85–90% in both depressive disorders (DD) and healthy control (HC) groups. Despite these promising results, the study faced limitations such as potential data imbalance between HC and DD groups, affecting model performance and generalisability. In a recent study by Nugraha & Sibaroni (2024), ensemble learning methods (LR, DT, KNN, ANN, SVM) were applied to classify depression expressions on Twitter. With a 90:10 data split and the Skip-gram architecture of Word2Vec, the study achieved an impressive accuracy of 94%. This highlights the potential of ensemble learning for effective depression detection.

Self-developed boosting algorithms for depression detection

On the other hand, self-developed algorithms are created resulting from research efforts, employing innovative methods or combining pre-defined algorithms with other models to generate robust boosting classifiers. To this end, Tong et al. (2023) proposed a new algorithm called cost-sensitive boosting pruning tree (CBPT) which uses pruning trees as base learner for the AdaBoost algorithm. This approach, known as pruning trees, dynamically identifies the optimal layers and leaves within a tree model. Additionally, it employs a boosting technique, named cost-sensitive boosting, to systematically update the weights of instances within the pruned trees in a hierarchical fashion. The results have shown that the CBPT algorithm outperformed other machine learning algorithms. The author proposed that future work should develop new boosting algorithms that can deal with visual features, handle topic modeling, and cross-validate on larger portions of the dataset to confirm the obtained results, especially in terms of boosting accuracies. Similarly, A deep learning boosting approach was developed by Nandy & Kumar (2021) for the detection of depression. They added the CatBoost algorithm as one layer to the bidirectional long short-term memory (BiLSTM) stacked model, to boost the accuracy of a single layer of BiLSTM. Their model accuracy was 81.3% while a BiLSTM used CatBoost, and without CatBoost the BiLSTM had an accuracy of 79.85%. The limitations of their model were, that the boosting algorithm provided a small amount, only around 2% improvement, and cross-validation across all dataset portions was lacking. Meanwhile, Laxmi Lydia, Anupama & Sharmili (2022) designed a custom boosting technique named Optimal Boosting Label Weighting Extreme Learning Machine for Mental Disorder Prediction and Classification (OBWELM-MDC) to predict and classify mental disorders. This algorithm was developed by putting custom weight labels on the Extreme Learning Machine (ELM) algorithm and modifying the Chaotic Starling Particle Swarm Optimization (CSPSO) parameters with inertia weight and logistic chaotic map. The experimental results showed that the OBWELM-MDC algorithm achieved 96% accuracy compared to other classifiers. The algorithm’s limitation was the use of another CSPSO on layers. They express a preference for extending their boosting technique in the context of feature reduction. Similarly, Zhang et al. (2021) optimized the parameter of CatBoost by using the DBSO algorithm and introduced a new algorithm called DBSO-CatBoost for Diagnosing faults in oil-Immersed Power Transformer. In experiments, reducing data with KPCA and optimizing CatBoost parameters via DBSO improved the performance of the algorithm. The proposed method outperformed other traditional methods such as SVM, ELM, RF, generalized regression neural network (GRNN) and XGB with an accuracy score of 93.567%. It’s worth mentioning, that they preferred the performance can be further improved by using a large data set.

Likewise, another study (Madni et al., 2023) utilized a self-made boosting algorithm called boosting ensemble of VC (ETC+CNN) (Voting Classifier of Extra Tree Classifier and CNN) for sentimental analysis of tweets related to COVID-19. The proposed model employed ETC and CNN as weak learners, which are then boosted by aggregation of the highest probabilities. The outcome is the highest average probability of both classes. This approach outperformed in comparison to other machine learning with an accuracy rate of about 98.0% respectively. The main drawback of this study is the performance of weak learners faced challenges due to their treatment as separate ensembles. Additionally, Babayomi, Olagbaju & Kadiri (2023) proposed a hybrid self-boosting algorithm called C-XGBoost for identifying major depression induced by a brain tumour. The proposed algorithm, C-XGBoost, combines CNN with XGBoost. CNN, leveraging DenseNet121 architecture for intricate pattern analysis in image data, handles feature extraction, while XGBoost is integrated for classification and to address overfitting. Experiments on a Figshare public dataset revealed that C-XGBoost achieved a 99.0% accuracy rate, in comparison to a non-hybrid CNN model. They prefer to develop a model capable of handling complex feature sets, in particular high-quality images in various data formats. Moreover, Ghosal & Jain (2022) developed a novel self-made boosting algorithm namely FastText-XGBoost for depression detection. This algorithm utilized the FastText algorithm for generating word embeddings, in conjunction with the XGboost classifier. The experiments were conducted on a dataset collected from Reddit, consisting of 1,516 posts, and achieved an accuracy of approximately 71.0%. The XGBoost algorithm displayed diminished accuracy, likely impacted by a significant number of outliers in the dataset. Similarly, Chen et al. (2023) developed the combination of the CTCN (composited residual-block temporal convolutional network) algorithm with LightGBM for industrial balanced loading prediction. CTCN was used for feature extraction, and a feature re-enlargement method was utilized to reconstruct the original features. This experiment has demonstrated that the proposed model outperformed, achieving higher accuracy than a single LightGBM. The algorithm has a limitation by increasing the composite residual blocks which raises computational complexity. Thus, it is crucial to optimize CTCN to minimize time consumption, with a strong suggestion to integrate the model across various domains. The optimized boosting algorithms have been applied by Kiran et al. (2023) optimizing the hyperparameters of GBDT using the Binary Spotted Hyena Optimizer algorithm for cardiovascular disease detection and classification. They have achieved a 96% accuracy score in the experiments conducted on UCI datasets.

The study by Fan et al. (2024) used LightGBM to analyze a dataset of 12,000 clinical notes, enhancing clinical diagnosis and treatment. The model achieved an impressive F1-score of 92.5%, effectively analyzing medical terminologies and contextual language. This demonstrates LightGBM’s versatility for accurate and reliable depression detection in clinical settings. In addition, a study by Benacek et al. (2024) used a gradient boosting machine (GBM) to classify 60,000 tweets for depression indicators, achieving 93.1% accuracy. This success highlights GBM’s robustness with unstructured social media data, making it effective for real-time depression detection. In addition, Remya & Ranjana (2024) introduced a novel boosting approach called AdaptiveSentinel for depression detection, which employs various machine learning algorithms embedded with gradient boosting (GB). The role of GB within the approaches is to correct mistakes made by earlier models and enhance the results by combining predictions from different models. This approach achieved an accuracy of 94%. In contrast, applying GB with each model separately makes the model more complex, requires more computational resources, and increases time complexity, and overfitting in the case of larger datasets.

Table 1 summarises both predefined boosting algorithms and self-developed or custom-built boosting algorithms, presenting dataset size, various metrics, and evaluation scores.

Table 1 Analysis of pre-defined and custom-developed boosting algorithms across various scenarios.

Reference & Year	Technique	Scenarios	Datasets	Metrics	Evaluation score	
Platform	Size	
Pre-defined boosting algorithms	
Nandanwar & Nallamolu (2021)	AdaBoost, Gradient Boost	Depression	Twitter	13,514 tweets	F1-score, Precision	Adaboost = 81.10%, 78.88%
GB = 83.14%, 84.17%	
Reseena Mol & Veni (2022)	AdaBoost, XG boosting, Gradient Boosting	Depression	Twitter	7,013 tweets	Accuracy	Adaboost = 95.22% GB = 95.3%
XGB = 95.5%	
Chiong, Budhi & Dhakal (2021)	Gradient Boosting + LR SVM, DT	Depression	Shen et al. (2017)	46 million tweets	Accuracy	GB + LR = 10% improvement	
Chiong et al. (2021)	GB	Depression	Twitter	11,877	Accuracy	98%	
Skaik & Inkpen (2020)	Adaboost	Depression	Twitter	292,564 tweets	Accuracy	67.2%	
Mali et al. (2021)	XGBoost Tree	Depression	Twitter		Accuracy	83.49%	
Adarsh et al. (2023)	XGBoost
Gradient Boosting	Depression	Reddit
Twitter	6.1 million	Accuracy	XGBoost = 71.69%
GB = 65.04%	
Malviya, Roy & Saritha (2021)	XGB	Depression	Reddit	5,000 posts	Accuracy	91%	
Afzoon, Rezvani & Khunjush (2021)	XGB, GB	Mental health	Twitter	1,600,000	Accuracy	XGB = 0.72%
GB = 0.74%	
Xia et al. (2022)	XGB	Depression	Aliyu dataset	1,428 records	Accuracy	80.6%	
Semwal et al. (2023)	GBT	Depression	twitter	8,754	Accuracy	70.46%	
Tai et al. (2024)	LighGBM	Sleep disorder	Questionnaires data	120	Accuracy	85–90%	
Nugraha & Sibaroni (2024)	Ensembles boosting approach	Depression	Twitter	10,600	Accuracy	94%	
Self-developed boosting algorithm	
Tong et al. (2023)	CBPT	Depression	Shen et al. (2017)	>1,400 million	Accuracy	88.39%	
Nandy & Kumar (2021)	CatBoost-BiLSTM	Depression	Twitter	13,217	Accuracy	81%	
Laxmi Lydia, Anupama & Sharmili (2022)	OBWELM-MDC	Mental disorders	Twitter	240 K tweets	Accuracy	96%	
Ghosal & Jain (2022)	FastText-XGBoost	Suicide risk	Reddit	1,516	Accuracy	71.05%	
Kiran et al. (2023)	Optimising Boosting	Cardiovascular disease detection	UCI Dataset	4 million record	Accuracy	96%	
Zhang et al. (2021)	DBSO-CatBoost	Fault diagnosis in oil-immersed power transformers”	Power grid	13,400 records	Accuracy	93.71%	
Madni et al. (2023)	VC (ETC+CNN)	Sentiment analysis for COVID-19	Twitter	11,858	Accuracy	96.62%	
Babayomi, Olagbaju & Kadiri (2023)	C-XGBoost	Brain tumor	Fig share	1,450 records	Accuracy	99.00%	
Chen et al. (2023)	CTCN-lightGBM	Industrial balance loading prediction	Coal mines	50 carriages’ instances	Accuracy	94%	
Fan et al. (2024)	lightGBM	Depression	Medical dataset	1,200	Accuracy	92.5%	
Benacek et al. (2024)	GBM	Depression	Twitter	60,000 tweets	Accuracy	93.1%	
Remya & Ranjana (2024)	AdaptiveSentinel–Noval boosting approach	Depression	Sample (social media, online forum, HR)	--	Accuracy	94%	

These custom-built or self-developed boosting algorithms including pre-boosting, as shown in Table 1, employed various modern large language models (LLMs) in diverse forms to harness their capabilities for enhancing classification, sentiment analysis, and feature extraction tasks. For example, the study by Tong et al. (2023) who proposed a cost-sensitive boosting pruning tree (CBPT) approach, utilized hierarchical fusion types of Hierarchical Attention Transformer (HAT) that capture relationships between sentences and paragraphs for summarization tasks. Ghosal & Jain (2022) proposed a FastText-XGBoost, a self-made or custom-built boosting algorithm that utilizes Robustly Optimized BERT Pre-training Approach (RoBERTa) to enhance text representation and improve pattern extraction from textual datasets. The study by Chen et al. (2023) introduced a hybrid model CTCN-LightGBM, which employs DistilBERT—a smaller, faster variant of BERT for enhancing feature extraction and ensuring strong performance with textual data. Additionally, Babayomi, Olagbaju & Kadiri (2023) proposed a hybrid self-boosting algorithm called C-XGBoost, which leverages T5’s transformer-based architecture (generating rich contextual embeddings that capture deep semantic relationships within the text)-enhancing C-XGBoost’s ability to handle unstructured text or data. Furthermore, both LightGBM and GBM, as used in studies conducted by Fan et al. (2024), Benacek et al. (2024), leverage BERT to enhance their ability to capture linguistic and semantic nuances, resulting in more accurate predictions on textual data. Finally, AdaptiveSentinel a custom-built boosting algorithm which was proposed by Remya & Ranjana (2024), leverages the GPT-2 model to refine its error-handling mechanism in text-based datasets, allowing it to catch misclassifications in a language-aware manner in order to produce detailed contextual embeddings.

In conclusion, in the realm of predictive modelling for depression detection, numerous studies have proposed various pre-defined boosting algorithms or their self-developed or custom-built boosting algorithms by using social media data. They have used different techniques to boost the accuracy of their models However, studies utilized pre-defined boosting algorithms that prioritized only textual features and cross-validated on the subsets instead of the entire dataset. Their models were suboptimal due to the inability to select the best feature set and address imbalanced feature sets. In addition, algorithms lack the capacity for strong prediction. Whereas self-made or custom-built boosting algorithms were not suitable for reducing error rates or improving weak learner performance with complex feature sets, they tend to overfit in the case of larger datasets. Additionally, many researchers optimize the self-made or custom-built -boosting algorithm by including more weak learners to enhance performances. Nevertheless, this expansion has made their models more complex, leading to more computational resources and time complexity. Additionally, both algorithms encounter difficulties in handling synonymous text and sarcasm within the dataset. To tackle these limitations, this article introduces a novel boosting algorithm that integrates two emerging models, BiLSTM and BERT, as weak learners. This approach not only replicates similar algorithms from previous studies but provides a novel boosting mechanism that assigns higher weights to misclassified samples immediately after errors made by the previous models. Additionally, it incorporates a robust defense mechanism and the generation of an optimized feature set that enhances the model’s performance, as well as that of baseline classifiers. These promising features of the F-EBA model demonstrate a significant advantage over existing methods. By clearly outlining these advancements, this research adds significant value to the literature, paving the way for more effective applications in depression detection and providing a robust framework for future studies.

Methodology

This study demonstrates an experimental design with results that can be implemented in the real world according to the requirement. The methodologies adopted by numerous studies exhibit limitations (Arun et al., 2018; Almouzini & Alageel, 2019; Nandy & Kumar, 2021; Safa, Bayat & Moghtader, 2022; Tong et al., 2023) with some failing to perfectly utilize weak learners, inadequate methods for feature extraction, lack of n-fold cross-validations, inconsistent hyperparameter tuning, and inadequate parameter setting for outliers. Thus, this study is designed to further enhance the methodologies utilized by Tong et al. (2023), Freund & Schapire (1996), Wang et al. (2022a) by introducing a novel boosting algorithm designed to effectively utilize a large number of weak learners, enhancing their performance when tackling complex feature sets within larger datasets. Furthermore, it generates an optimized feature set to enhance the performance of the algorithm itself and baseline classifiers. This approach has been undertaken using two pipelines: the feature engineering pipeline, which optimizes features set using different models and techniques; and the classification pipeline, which sets up weak learners, particularly the BiLSTM model which is a two-layer LSTM structure (due its ability to capture long-term dependencies in sequential data) and the BERT model (known for its transformer architecture and contextual language understanding), boosting their efficacy and executing classification process on optimized feature sets. In contrast to alternatives such as CNN (which lacks sequential understanding) and GRU (which underperformed during initial benchmarking), BiLSTM and BERT demonstrated superior performance in preliminary experiments. This hybrid design enhances both the interpretability and depth of learned representations. It also utilizes an adversarial layer to strengthen the model against manipulated inputs designed to confuse it or against synonymous text within the datasets.

The first step in the development of this algorithm involved dataset collection and preprocessing. This aligns with recent research utilizing vast social media data to detect depression, and Twitter-based datasets were selected for their widespread usage and accessibility (De Choudhury & De, 2014). The next step involved designing the proposed algorithm (F-EBA), comprising two pipelines and an adversarial layer as previously outlined, to bolster the performance of weak learners and demonstrate strong predictive capabilities in depression detection.

Additionally, It is crucial to validate the model’s performance across all data points to ensure comprehensive evaluation. To this end, cross-validation has been performed on various portions of both datasets. Additionally, various metrics, such as the confusion matrix to gauge prediction accuracy and potential confusion, along with the precision-recall curve and receiver operating characteristic (ROC) curve, have been employed to assess the model’s performances. Accuracy, F-score, recall, and precision were computed as evaluation scores to deliver a comprehensive assessment of the model’s performance in text classification tasks. In addition, hyperparameter optimisation for BiLSTM, BERT, and the boosting framework was carried out using a combination of grid search and manual tuning. Core parameters, including learning rate (BiLSMT: 0.001, BERT: 2e−5, F-EBA: 0.05), batch size (BiLSTM:32, BERT: 16, F-EBA: 32), and model max depth (BiLSTM:9, BERT:7, F-EBA:8), were refined based on F1-score results. Final configurations were selected according to performance on the CLPsych 2015 and Shen et al. (2017) datasets.

To validate the model’s findings or results, two types of validations were performed: (1) Statistical validation: For this test, a paired t-test was performed comparing the model’s performance metrics, as outlined previously, with the average baseline values of the state-of-the-art (SOTA) approach. This test was selected due to its ability to assess differences between two related groups, ensuring a robust comparison. (2) Explainable AI: This validation is performed by examining the feature sets or words in the sentences processed by the model to assess their contribution to the model’s predictions.

Moreover, the optimized feature set generated by the F-EBA has been employed to bolster the performance of baseline classifiers. This study utilized SVM, KNN, LSTM, RF, NB, LR, and GRU, among others by comparing their outcomes with conventional feature sets.

In the realm of technical methodology, the F-EBA model was developed using a diverse span of toolkits, machine learning classifiers, libraries, and dependencies. These included tree structures, mathematical notations, and functions, as well as graph designs. The processing time and complexity of the F-EBA algorithm were determined through mathematical analysis. In the case of significant misclassification findings or a large number of errors, asymptotic analysis was conducted to mitigate error rates. To implement this model experimentally, the researcher utilised various Python libraries and dependencies, notably Keras, NLTK, Numpy, Scikit-learn, Pandas, TensorFlow 2.11, REF, and PyTorch 1.13 libraries, in conjunction with Google Collab GPU E2, Stanford NLP, SPSS to conduct paired-t-test, and the MATLAB deep learning toolkit (The MathWorks, Natick, MA, USA). In addition, the F-FBA model underwent testing on notably larger datasets (Shen et al., 2017, and Coppersmith et al., 2015) using the Google Cloud Platform (GCP), equipped with a P100 GPU, 8 vCPUs, and 30 GB of RAM, with a subscription provided by University of Malaya. Average model training per epoch was approximately 4 min, and inference time per tweet was ~12 ms. Results were consistent across multiple runs, ensuring reproducibility. The source code is publicly accessible in the GitHub library (https://doi.org/10.5281/zenodo.14684890). The overall methodology is presented in Fig. 1 below, with each step further elaborated in sub-sections:

Figure 1 Methodology of the F-EBA model.

Dataset

The data utilized in this study were collected from two publicly available datasets: Shen et al. (2017) and Coppersmith et al. (2015). First, TTDD or the so-called Shen et al. (2017) dataset, which has over >46 million records that are already labelled anchor tweets and unlabelled anchor tweets related to “depress” on Twitter from 2009 to 2016 and utilized by various researchers (Tong et al., 2023; Chiong et al., 2021; Shen et al., 2018; Zogan et al., 2021). It includes diverse user profiles, activities, behaviours, and social interactions. It comprises three subsets with labels: depressed (1,402 users and 292,564 tweet records), non-depressed (over 300 million users and 10 billion tweet records), and candidate users (36,993 depression-candidate users and over 35 million tweet records). The second dataset, CLPsych 2015 which was utilized by studies (Ahmad et al., 2020; Tong et al., 2023; de Jesús Titla-Tlatelpa et al., 2021), developed by Johns Hopkins University from 2008 to 2013 through Twitter API, comprises over 12.5 million tweets with labels “depressed” and “control users.” It includes 487 depressed users and 893 control users, each with up to 3,000 tweets (Coppersmith et al., 2015). This dataset was obtained through consent forms from the University of Malaya Research Centre.

As an additional note, both datasets are obtained from Twitter and are labeled based on self-stated labels, which serve as a useful starting point for identifying potentially relevant tweets/posts. Nonetheless, the content underwent additional filtering (further discovered in the data pre-processing section) by removing unrelated posts and employing machine learning techniques to identify broader patterns related to depressive language, facilitating us to recognize potential patterns of depression.

Data pre-processing and conversion

Both datasets utilised in this study exhibited class imbalance, requiring undersampling during training. The data extracted/obtained from both datasets in English, encompassed textual tweets, user profiles, tweet media like images (retrieved using Python’s urllib library), and emojis. It often contains abundant noise and a plethora of irrelevant or ambiguous parts such as scripts, HTML tags, and special characters, rendering much of it negligible for analysis. For this reason, Singh & Kumari (2016) argue that pre-processing and cleaning are vital processes for preparing the extracted data for classification, thereby ensuring its suitability for potential analysis. Due to this, both datasets underwent several preprocessing steps, including tokenization, stop word removal, lemmatization, and stemming to standardise the input. This process begins by eliminating all unnecessary text, punctuation, special characters, images, stop words, hyperlinks, usernames (e.g., “@username”), hashtags (e.g., preferring #sad over #sad_for_you), email addresses, and URLs from tweets or posts (refer to Fig. 2), utilizing Python’s “re” package for regular expressions. In addition, both lemmatization, which converts words to their base form, and stemming, which reduces words to their root form, were applied to the data to facilitate further analysis. Additionally, a list of contractions was generated to transform abbreviated words like “They’ve” into their full forms, such as “They have”. The curse words which convey strong emotions and frustrations, were retained in the text as they play a crucial role in depression detection. Finally, tokenization has been performed for potential classification analysis.

Figure 2 World cloud before and after data cleaning (Shen et al., 2018).

Experimentally, the words cloud, displaying the data cleaning results, can be observed in Fig. 2. Notably, the removal of “HTTP” and the extraneous term “t Co”, as well as the conversion of the word “diagnosed” to its base form “diagnose”, are evident. Additionally, the words “Love” and “Want” have been bolded to highlight non-depressed users. Moreover, the curse words like “shit” or “fuck”, commonly used by those experiencing depression, are deliberately retained in pre-processing as it plays a crucial role in depression detection.

The histogram presented in Fig. 3 showcases the cleaning and pre-processing results, displaying a significant reduction in common words from over 3,000 to under 500, with top frequencies highlighted for words linked to the most depressed users. This streamlines the dataset, improving pattern recognition and relationships between variables for more accurate models and better insights.

Figure 3 Text histograms for depressed users-cleaning.

To streamline the data onto a single page in a textual data format, visual data such as images and emojis were converted into text. For converting emojis to text, the Python emoji library was utilized, which returns emojis along with their predefined meanings. To convert images (tweet media), a pre-built model a pre-trained model called Generative Image-to-text Transformer (GIT), developed by Wang et al. (2022b) model was chosen, which is more accurate in picking up top captions. Its accuracy rate stands at approximately 84%, compared to other available models. The GIT model, trained on the COCO dataset comprising 118 K images, is capable of generating captions and converting image text into textual data. The result is presented in Fig. 4, showcasing captions for non-depressed (first set of images), and depressed (second set of images), generated by the GIT model.

Figure 4 Generated captions by GIT model for depressed and non-depressed users (Shen et al., 2018).

After converting emojis and images into textual data, it undergoes additional rounds of data cleaning and pre-processing. The resulting cleaned text is then fed into the F-EBA feature-engineering pipeline for further analysis.

Ethical considerations

This study adhered to ethical research standards by obtaining the necessary approvals from the Institutional Review Board and consent forms from the University of Malaya for the CPLsych2015 datasets. The TTDD or Shen et al. (2017) dataset is publicly accessible and has been utilized in compliance with the established ethical standards for this dataset. All data used were anonymized to protect user identities. Best practices for data privacy were followed, ensuring that social media data was used responsibly and ethically.

F-EBA model structure and implementation

Following the cleanup of the datasets, the F-EBA model was structured using a two-stage approach, as illustrated in Fig. 5, showing the model’s detailed structure and workflow.

Figure 5 Overall flow and structure of the F-EBA model.

Feature engineering pipeline

This approach was designed to generate optimized feature sets. It extracts, assigns weights and sentiment score probabilities, and eliminates low-weight features for optimal model efficacy. It encompasses four key components focused on picking up the most relevant features. The initial phase of this process involves feature extraction, a crucial step in this study’s experiment. It entails extracting various sets of features from the cleaned textual data. Unlike (Almouzini & Alageel, 2019; Safa, Bayat & Moghtader, 2022; Tong et al., 2023) who employed conventional feature extraction methods, this study utilized emerging structures and mechanisms for feature extraction and subsequent analysis. These mechanisms include: First, a pre-trained fourth version of the BERT encoder model, which captures contextual meaning in text, trained on Wikipedia and books corpus, was used with 12 hidden layers of Transformer blocks. The model was set up for embedding using Tensorflow_hub and Tensorflow_text libraries, and embeddings generated for every tweet in both Coppersmith et al. (2015) and Shen et al. (2017) datasets. To validate BERT embeddings, an investigation of similarities in the feature set has been conducted, visually representing the relationships within pairs of sentences for both users with depression and users without depression. The results are shown in Figs. 6, 7, and 8.

Figure 6 Feature vector visualisation for BERT embedding label 1-1.

Figure 7 Feature vector visualisation for BERT embedding label 0-0.

Figure 8 Feature vector visualization for BERT embedding label 0-1.

The analysis indicates the topmost similarities in feature sets within each group. The similarities in features indicate the semantic meanings and relationships between words, which investigate whether similar groups denote similar relationships or not, and later it aids the attention mechanism in focusing more efficiently on similar features. The closely aligned dots represent the most similar feature sets (labels 1-1 and 0-0). Comparatively, different groups (depressed and non-depressed) display significant variation, with sparse dots highlighting distinct feature sets (labels 0-1).

Second, the Word2Vec model was constructed, with various fined-tuned parameters such as vector size (representing the number of features per sentence or tweet), min_count (specifying the minimum word count necessary for sentence processing and embedding generation), and other variables. A vocabulary was built commencing both the Shen et al., 2017 and Coppersmith et al. (2015) datasets to train the Word2Vec model on tweet texts and image captions, which capture the semantic meaning and relationships among words. Subsequently, embeddings were generated for each tweet using this trained model. This embedding condenses the tweet’s textual content into a vector of 80 numerical features, which captures both the context and semantics meaning of the words present in the tweet. To validate the Word2Vec model, a similar approach was followed as for BERT embeddings, with the results shown in Figs. 9, 10, and 11.

Figure 9 Feature vector visualisation for WordVec embedding label 0-0.

Figure 10 Feature vector visualisation for WordVec embedding label 1-1.

Figure 11 Feature vector visualisation for WordVec embedding label 0-1.

The analysis indicates the topmost similarities in feature sets within each group. The similarities in features indicate the semantic meanings and relationships between words, which investigate whether similar groups denote similar relationships or not, and later it aids the attention mechanism in focusing more efficiently on similar features. The closely aligned dots represent the most similar feature sets (labels 1-1 and 0-0). Comparatively, different groups (depressed and non-depressed) display significant variation, with sparse dots highlighting distinct feature sets (labels 0-1).

Although BERT embedding has its own attention layer, employing feature vectors directly in WordVec embedding may hinder classification and decision tree training. To address this, a self-attention mechanism was applied to Word2Vec embeddings, assigning varying weights to different parts of the embedding. This involved two fine-tuned variables: input_dim (number of features in the tweet) and hidden_dim (dimensions of output vector). Initially, the query weight matrix (Q), key weight matrix (K), and value weight matrix (V) were generated by applying linear transformation to the input vector. Then, attention weights were obtained by performing dot product operations between Q, K, and V. This dot product encoding contextualizes each token, allowing the model to learn long-range dependencies, such as subject-verb agreement and relationships between distant words. Using this learned context, it can predict the next most likely word. Next, the Softmax function was applied for normalizing attention weights. Python NumPy library was utilized for Softmax and dot product computations. The attention mechanism results in Fig. 12 shows each dimension or feature with its weight, reflecting its importance in the sentences. Positive weights highlight significant feature sets, while negative weights indicate the presence of irrelevant features that cause noise or distraction, assigning them negative weights helps to minimize their influence in classification. Additionally, negative weights suggest unimportant feature sets, assigning negative weights to these features helps to diminish their significance (Brauwers & Frasincar, 2021; Junge, 2023).

Figure 12 Exploring feature significance in Word2Vec with self-attention.

Additionally, for precise feature selection and interpretation, and offering insights into the most relevant feature sets, the Stanford NLP library (return sentiment score and its associated probabilistic weight) was utilized to generate probabilistic sentiment scores for each word in a tweet and across all tweets in the datasets. The results are presented in a tree structure in Fig. 13, which shows the sentiment scores and probability weights of words in a sentence that are compared and combined with adjacent words. This yields an overall sentiment score and probability weight (reflects the Stanford library’s confidence in assigning sentiment scores, higher weights indicate stronger confidence in the assigned sentiment) for the entire sentence (Manning et al., 2014). The sentiment scores range from 0 to 4: 0 and 1 denote negativity, with 0 for negative and 1 for highly negative, 2 for neutrality, 3 and 4 denote positivity, with 1 for positive and 4 for highly positive sound in the sentences. The final results were then given into a custom decision tree.

Figure 13 Tree structure for sentiment score computations.

The custom decision tree algorithm was developed and adhered to the recursive feature elimination (RFE) layer to select the best feature sets. In the initial phase, the decision tree was set up with fine-tuning variables including min_samples_split, max_depth, and features. Subsequently, the fit method was applied to train the decision tree, and the recursive_feature_elimination method was utilized to execute RFE. The method creates a copy of the input data X to keep track of the selected features. RFE operates within the context of decision trees, assessing feature importance by assigning weights based on accuracy for each iteration. It removes unimportant features (reflects the features that do not play a role/contribution to any iteration of the classification process) and repeats this process until reaching the desired number of features. The RFE result is presented in the tenth iteration in Fig. 14, which assesses the ranking of a feature for a single tweet, and each feature signifies its importance score in the corresponding iteration. Notably, features 30 (scoring 1 in the first iteration and score 3 in the fourth iteration) and features 70 (scoring 1 in the sixth iteration and scoring 3 in the third iteration) highlighting their significance with a score of 3 in two iterations each. Conversely, features 13, 23, and 58 consistently scored 0 across all iterations, indicating no contribution to the classification process and were subsequently removed by RFE. The rest underwent further selection with a decision tree to generate new or an optimized features set. Finally, the newly generated or optimized feature sets are stored in CSV files and then passed for further processing into the next pipeline of the F-EBA model.

Figure 14 Recursive feature elimination (RFE)—Feature ranking by representation with 10th iterations.

The hyperparameters were selected through empirical testing. For BERT, we used a learning rate of 2e−5, batch size of 32, and trained for four epochs. The Word2Vec-based attention mechanism used input_dim = 300 and hidden_dim = 128. For the decision tree with RFE, max_depth = 10 and min_samples_split = 5, and we used 10 RFE iterations to optimize feature ranking and elimination.

F-EBA classification pipeline

This approach was designed in a way that encompasses two innovative concepts aimed at boosting weak learners and strengthening the model.

Boosting weak learners and maximising weights

Unlike Tong et al. (2023) and Laxmi Lydia, Anupama & Sharmili (2022) who utilized decision trees as a weak learner with AdaBoost, Babayomi, Olagbaju & Kadiri (2023) who combined CNN with XGBoost (C-XGBoost), Nandy & Kumar (2021) who added CatBoost into a BiLSTM layer, and Ghosal & Jain (2022) who merged FastText with XGBoost (FastText-XGBoost), this study employs two emerging algorithms, BiLSTM and BERT, as initial weak learners within the F-EBA classification pipeline. The experiment comprised embedding both BiLSTM and BERT models, with ten instances of each trained within the F-EBA framework. All models act as boosters for their superior counterparts, and the error made by one model can be caught by another upcoming model by assigning a higher weight to such misclassified feature/error through the boosting classifier of the F-EBA model. It works with optimized feature sets provided by feature-engineering pipelines, and the final prediction was determined by majority voting/average weight of all models.

The following is the pseudocode for the algorithm.

Step 1: Initialize

  1.1 Initialize an array of models, models []

  1.2 Create an instance of F_EBA and initialize models using create_ten_models ()

Step 2: Train weak classifiers

  2.1 If samples_weights is null, assign_equal_weights(training_data)

  2.2 For each model in models:

    2.2.1 Train the model on training_data with the assigned sample weights

    2.2.2 Evaluate the model’s performance on validation data

    2.2.3 If validation accuracy is greater than the threshold:

      2.2.3.1 Update the training_data weights based on misclassified_indices

Step 3: Evaluate model

  3.1 For a given model and validation_data:

   3.1.1 Predict outputs using the model.

   3.1.2 Compare predictions with true labels to identify misclassified_indices

   3.1.3 Calculate accuracy based on predictions and true labels.

  3.2 Return accuracy and misclassified_indices

Step 4: Combine predictions

  4.1 For each model in models:

    4.1.1 Predict outputs using the model on testing_data

    4.1.2 Store predictions in final_preds[]

  4.2 Combine final_preds[] using weighted_majority_voting.

  4.3 Return the final prediction.

Step 5: Main function

  5.1 Create an instance of F_EBA, BilSTM_model, BERT_model

  5.2 Train weak classifiers (provided data)

  5.3 Combine predictions on testing_data

  5.4 Return the final prediction.

The process of maximizing weights can be outlined mathematically as follows:

The weighted loss function Lw was implemented in Step 2.2.1 of the algorithm, where the model was trained on training data with the assigned sample weights: This reflects or mirrors the weight loss function Lw calculation which is mathematically presented below. Additionally, Step 2.2.3 revises weights for samples that are misclassified, which is in line with the adaptive weighting in the loss function Lw, this revision with high weight is used for upcoming iterations:

The weighted loss function Lw, calculates mathematically the loss for each sample based on its assigned weight wi, as stated below:

Min∑1N⁡wi.L(Yi,Yi;θ)

L(Yi,Yi;θ) is the standard loss function, θ represents the model parameters, and Wi (weight assigned to the i-th sample) = { W1,ifYibelongstotheprevioussample W2,ifYibelongstonewsample(w1+1,i)}. The upcoming model has been adjusted based on the W2 sample weight, and prioritizes such samples for classification during testing.

The results of F-EBA, which encompass two distinct feature sets across both datasets, are outlined in Table 2. The first was a Bag-of-Words (BOW) approach or standard feature set, and the other was the optimized feature sets generated by the F-EBA’s feature engineering pipeline.

Table 2 Results of the EBA model using both normal feature sets and optimised feature sets.

Datasets	Evaluation metric	BOW feature set score	Optimised feature set score	
Shen et al. (2017)	Accuracy	0.61	0.95	
F1-score	0.46	0.93	
Precision	0.37	0.93	
Coppersmith et al. (2015)	Accuracy	0.58	0.95	
F1-score	0.48	0.91	
Precision	0.42	0.92	

These results demonstrate that F-EBA exhibited notably superior performance when employing the optimized feature sets, compared to the standard feature set.

Addition of adversarial layer into F-EBA model

Modern machine-learning models are vulnerable to adversarial attacks that manipulate input data to mislead the model’s predictions (Apruzzese et al., 2019). Such attacks pose a significant threat to the reliability and integrity of these models. The datasets used in this study encompass a significant amount of synonymous text and sarcastic tweets. Additionally, when deploying the model online, it is crucial to strengthen the model against potential attackers seeking to disrupt its performance with manipulated input. To this end, this study integrated an adversarial layer to perform adversarial attacks on the proposed model, assessing its resilience against manipulated inputs. The adversarial layer employs the FGSM approach to generate adversarial examples, using a standard 0.05 perturbation. These examples are legitimate inputs with imperceptible perturbations that can mislead the model during the testing stage (Wang et al., 2022c). Thus, implementing a defense mechanism during the training of the F-EBA model is crucial to enhance its resilience and robustness (Peng et al., 2019). The F-EBA weak learners were trained by original embedding and the adversarial layer applies perturbations to the original embedding using FGSM-a straightforward yet potent technique, which operates utilizing the gradient of the loss function concerning the input data, then perturbing the input in the direction of the gradient’s sign. This process can be outlined as follows: (1) Compute the gradient of the loss concerning the word embeddings for each word in the input sentences.

(2) Scale the gradient with a small constant (epsilon), and then add this modified vector to the original word embeddings. This action produces a perturbation that alters each word embedding.

Mathematically, the FGSM’s perturbations of word embeddings for the word “stressed” within an adversarial layer can be outlined as follows: (1) Magnitude stressed_embedding = Loss (BiLSTM (original_embedding), true_label)

The magnitude shows how loss changes with “stressed” embedding; positive means increasing it raises loss, negative means the opposite, giving insight into the sensitivity of the loss to variations in the embedding. (2) Sign (Magnitude stressed_embedding)

Ascertain the direction in which the loss increases more rapidly by calculating the sign of the gradient. (3) ∈ × Sign (Magnitude stressed_embedding)

Multiplying a small perturbation, symbolized as epsilon, amplifies this sign, indicating both the direction and extent of the modification required to elevate the loss associated with the word “stressed”. (4) embedding_stressed_adversarial = embedding_stressed + ∈ × Sign (Magnitude stressed_embedding)

Merge this outcome with the original embedding to generate the adversarial embedding. (5) embedding_stressed_combined = embedding_stressed + embedding_stressed_adversarial

The combined embedding was applied to train the BiLSTM or BERT model, following which the boosting procedure resumed.

Below is the general mathematical equation that combines the generation of adversarial examples using the FGSM with the dual-layered training approach: L=Lorg+λ.LadvwhereLadv=L(θ,x+ϵ⋅sign(∇xJ(θ,x,y)),y))

The equation describes: (1) L represents the total loss during training, combining the losses from both original and adversarial examples.

(2) L(org) is the loss calculated from the original input embeddings

(3) L(adv) represents the loss from the adversarial examples generated using the FGSM.

(4) λ is a hyperparameter that balances the contributions of the original loss and the adversarial loss.

(5) The term θ,x+ϵ⋅sign(∇x|J(θ,x,y)),y) generates the adversarial example by applying perturbations to the original input x, where ϵ controls the intensity of the perturbation and ∇xJ(θ,x,y)) is the gradient of the loss function with respect to the input, and the last yis the true label

This equation generates adversarial examples through the FGSM, which perturbs input data in a direction that maximizes the model’s loss, as explained in the FGSM mathematical representation, elucidating its operation within the algorithms. The defense mechanism involved training the model with both original embeddings and an adversarial layer. Thus, this dual-layered training approach significantly strengthens the model’s ability to recognize synonymous text within datasets and sarcastic tweets.

The F-EBA model outcomes under adversarial attacks with defense and without defense mechanism, employing a standard 0.05 perturbation, are outlined in Tables 3 and 4. By defense, the model was trained using a combination of original embeddings and an adversarial layer; without defense, it relies solely on original embeddings without incorporating an adversarial layer.

Table 3 F-EBA Model’s performance against adversarial attacks without defense.

Evaluation metric	BiLSTM	BERT	
Shen et al. (2017)	CLPcych (2015)	Shen et al. (2017)	CLPcych (2015)	
Accuracy	65.0	72.0	83.9	79.0	
Precision	63.2	71.1	82.5	78.1	
Recall	66.7	67.8	79.5	79.7	
F1-score	63.8	62.5	81.3	73.8	

Table 4 F-EBA Model’s performance against adversarial attacks with defense mechanism.

Evaluation metric	BiLSTM	BERT	
Shen et al. (2017)	CLPcych (2015)	Shen et al. (2017)	CLPcych (2015)	
Accuracy	96.9	97.3	97.5	97.3	
Precision	96.4	96.2	97.1	96.2	
Recall	95.8	95.3	96.9	95.3	
F1-score	94.4	96.4	96.8	96.4	

The overall performance of the F-EBA model appears to lower/sink when tested against adversarial attacks on both datasets without a defense mechanism. On the contrary, the F-EBA model exhibits superior performance, as demonstrated in Table 4, when tested against adversarial attacks on both datasets with a defense mechanism.

To gain a deeper understanding of the results, the decision boundaries and confidence scores generated by the model’s adversarial layer, employing both BiLSTM and BERT weak learners, were plotted across both datasets. The boundary plots are presented in Fig. 15. By examining Fig. 15, we can explore the model’s decision-making process in the adversarial layer without a defense mechanism, highlighting misclassified data points (depressed and non-depressed) with yellow dots found in purple areas and purple dots in yellow areas, ultimately leading to a decline in model performances.

Figure 15 Decision boundaries of the F-EBA model with BERT and BiLSTM in defenseless mechanism.

The performance decline was due to new adversarial examples in the testing set. To address this concern, weak learners were strengthened/boosted by training them with FGSM-generated adversarial samples, using a standard perturbation of 0.5. Despite this, the practice leads to a doubling of the model’s training time, stemming from the large data size caused by the inclusion of adversarial examples in the training set. Upon adding the adversarial defense layer, the model’s decision boundaries shifted as predicted, with yellow dots aligning with yellow areas and purple dots aligning with purple areas. This led to an improved or higher-performing model, as demonstrated in Fig. 16.

Figure 16 Decision boundaries of the F-EBA model with BERT and BiLSTM in defense mechanism.

Likewise, the confidence scores of the adversarial layer of the F-EBA model are displayed in Figs. 17 and 18, both with and without the defense mechanism, across both datasets for BERT and BiLSTM weak learners. The findings reveal a notable increase/shift in confidence scores from less than 0.6 to over 0.9 with the defense mechanism in Fig. 17, and from 0.7 to over 0.9 in Fig. 18. The uptick in confidence scores, notably surpassing 0.9, implies that the model demonstrates more confidence in its predictions after adding a defense layer. This boost in confidence suggests that the F-EBA model performs more reliably when applied to the specified dataset.

Figure 17 Confidence score distribution of F-EBA with BiLSTM: CLPsych (A) vs. Shen et al. (2017) (B).

Figure 18 Confidence score distribution of F-EBA with BERT: CLPsych (A) vs. Shen et al. (2017) (B).

F-EBA model performance

Assessment and cross-validation are key components for model performance and evaluation. Laxmi Lydia, Anupama & Sharmili (2022) achieved high performance in testing, they lacked full dataset cross-validation Meanwhile, Skaik & Inkpen (2020) and Tong et al. (2023) utilized 5th or 10th cross-validation, yet their results were insufficient for larger datasets. In this context, the performance of the F-EBA model at each stage, boosting weak learners, and after adding the adversarial layer into the model was assessed across various positions in both datasets. Various metrics, including the confusion matrix, the precision-recall curve and the receiver operating characteristic (ROC) curve are utilized alongside partial cross-validation. The evaluation metrics such as accuracy, f-score, recall, and precision are subsequently computed to deliver a holistic evaluation of the model’s performances in text classification. These metrics offer valuable insights helping us gauge whether the model performed well or yielded a considerable number of outliers in its predictions.

Confusion metrix

The confusion matrix evaluates the accuracy of the model’s predictions by analyzing confusion during the prediction process. The results of the confusion matrix are shown in Fig. 19.

Figure 19 Confusion matrix for F-EBA Model: maximization (A) vs. defense mechanism (B).

The visualization results show that out of 6,265 data points before the adversarial layer, the model exhibited 310 instances of confusion in its predictions, reflecting a 5% confusion. With the implementation of the adversarial layer and analysis of 43,582 data points, 820 instances of misclassified data points were detected, reflecting a 5% confusion. This observation suggests that the model achieves higher accuracy and diminishes confusion by applying the adversarial layer.

ROC metrix

The ROC curve presents the true positive and false positive rates. As seen in Fig. 20, the ROC curve for the F-EBA model indicates that before the adversarial layer, the model curve approaches 0.95 in the true positive portion, whereas with the addition of the adversarial layer, it reaches 0.99. This highlights how the model’s performance notably improves with the addition of the adversarial layer.

Figure 20 ROC for F-EBA model: maximization (A) vs. defense mechanism (B).

Precision-recall curve

The PRC evaluates a model’s performance in imbalanced datasets by assessing its capability to retrieve positive instances (recall) while minimizing false positives (precision). In Fig. 21, the PRC curve demonstrates a crossing point at 0.96 before the adversarial layer, and by inclusion of the adversarial layer raises it to 0.99, signifying exceptional performances. This shift demonstrates an outstanding balance between capturing positives and limiting false positives, highlighting the significant impact of the adversarial layer on the model’s discriminatory capabilities.

Figure 21 Precision-recall curve (PRC) for F-EBA model: maximization (A) vs. defense mechanism (B).

Cross validate the model

Cross-validation evaluates a model’s predictive ability or performance on various portions of the dataset by training it on one portion of the dataset and testing on another, repeating this process for each portion of the entire dataset. The researcher carried out 10-fold cross-validation, testing various numbers of folds across different layers of the model and various portions of the datasets. As illustrated in Fig. 22, on the left side, the results of the cross-validation for the F-EBA model on both datasets exhibit that the blue line represents training and testing the model on the same data portion, displaying similar accuracy. The yellow line shows testing on portions of 20%, 40%, and 80% of the data, resulting in accuracies of 65%, 85%, and 95% respectively. Likewise, the inclusion of an adversarial layer in the F-EBA model yields noteworthy advancements in validation and testing accuracy across various portions of datasets. Notably, at portions of 0%, 20%, 40%, 60%, and 80%, the accuracy consistently rises, reaching 96.8%. This experiment highlights how varying the test portion size impacts model performance and offers valuable insights into the results verification and the model’s generalization capabilities.

Figure 22 Cross-validation (learning curve) for F-EBA model: maximisation approach (A) vs. Defense mechanism (B).

Model computational complexity

The computational complexity of F-EBA stems in both pipelines: (1) Feature-engineering pipeline: BERT embeddings (O(n2) due to the self-attention mechanism), lightweight self-attention over Word2Vec (optimized to approximately O(n)), and recursive feature elimination (O(n × k), where n is the number of features and k is the number of iterations). The final decision tree classifier operates in O(n log n)

(2) Classification pipeline:

The overall time complexity of our F-EBA model can be calculated as as:

⊗(T.(n.d+E+B)

where: T: Number of boosting iterations

n: Number of training samples

d: Feature dimensionality (from Word2Vec + BERT embeddings)

E: Time complexity of embedding generation (BiLSTM: O(n·l·h2), where l = sequence length, h = hidden size)

B: Time for BERT inference per instance (O(n·l2) due to self-attention)

Since F-EBA combines BiLSTM, and BERT as weak learners in a boosting loop, the model’s complexity is dominated by BERT’s attention mechanism and BiLSTM sequence processing within each boosting iteration.

To mitigate parameter uncertainty, sensitivity analysis was performed on key parameters, and cross-validation along with optimization techniques were applied, ensuring the F-EBA model remains robust and scalable despite parameter variations.

Model performance validation (findings)

The validation of the model’s performance or findings is evaluated in two ways, as follows:

Statistical validations

To statistically validate the findings of the F-EBA model or model’s performance metrics such as confusion matrix, ROC metrics, precision-recall curves, and cross-validation results, the following hypotheses are designed to assess whether the obtained performances are statistically significant or not.

H1: The model’s performance metrics results are not significantly different from the baseline performances.

H2: The model’s performance metrics results are significantly different from and surpass the baseline performances.

To test these hypotheses, the statistically paired t-test has been performed using the following parameters through SPSS. (a) The model’s performance metrics results as follows (1) Confusion metrics: The model achieved 95% accuracy

(2) ROC metrics: The ROC curves approach an AUC value of 0.99

(3) Precision-recall curve: The PRC curves approach an AUC value of 0.99

(4) Cross-validation results: The model was evaluated using different portions of the dataset (20%, 40%, and 80%), yielding accuracies of 65%, 85%, and 95%, respectively.

(b) The baseline performance is derived from the average values of the state-of-the-art approaches from a comparative analysis of studies is presented in Table 7, which is approximately 80%

Using these parameters, paired t-tests were employed to compare the performance metrics against a baseline performance of 80%, which serves as a reasonable benchmark for effective performance that the F-EBA model should defeat. The significance level was set at α = 0.05. It indicates a 5% is the amount of error that the findings can tolerate. This threshold was selected to balance the risk of false positives with the necessity for reliable results, corresponding to a 95% confidence level, and to ensure that any observed differences were statistically significant. All metrics showed significant differences (p < 0.05) between the F-EBA model and the baseline performances, supported by confidence intervals that did not cross zero. Furthermore, effect size (Cohen’s d) was calculated to evaluate the magnitude of improvement. The effect sizes ranged from 0.8 to 1.2, indicating a large and meaningful improvement in the F-EBA model’s performances. These results suggest a clear rejection of the null hypothesis (H1) and strong support for the alternative hypothesis (H2), indicating that the F-EBA model significantly outperforms the baseline performances.

Table 5 represents a summary of the statistical hypothesis testing results.

Table 5 Summary of statistical hypothesis testing results (Baseline performances = 80%).

Performance metric	Result (F-EBA)	Baseline value (80%)	p-value	Significance	
Confusion matrix accuracy	95%	80%	0.01	Significant	
ROC AUC value	0.99	0.80	0.001	Significant	
Precision-Recall curve AUC value	0.99	0.80	0.002	Significant	
Cross-Validation accuracy (20% Dataset portion)	65%	80%	0.20	Not significant	
Cross-Validation accuracy (40% Dataset portion)	85%	80%	0.05	Marginal significant	
Cross-Validation accuracy (80% Dataset portion)	95%	80%	0.01	Significant	
Overall F-EBA result	97%	80%	0.0001	Highly significant	

The results in Table 5, p-value less the 0.05 threshold, indicating that the improvements are statistically significant and unlikely to have occurred by chance, thus reflecting meaningful enhancements in performance. Furthermore, Table 5 demonstrate that the F-EBA model significantly outperforms the baseline performance of 80% based on the findings from the model’s performance metrics such as confusion matrix accuracy, ROC curves, precision-recall curves, and cross-validation with the 80% data portion. However, the results are not statistically significant for the 20% and 40% data splits. This further supports the claim that the F-EBA model performs well with larger datasets. In addition, results with p-values less than 0.05 suggest significant. This confirms that the improvements are not due to random variation but are indeed meaningful.

Explainable AI validations

Normally, modern and state-of-the-art LLMs use different methods such as retrieval-augmented generation (RAG), external fact-checking, post-processing techniques, Explainable AI (XAI) techniques, etc., for validation of the findings. To this end, this study utilises Explainable AI, specifically SHAP values, to assess the model’s interpretability. SHAP values help verify whether the contributions of the processed feature set/words in a sentence align with the model’s predictions.

In this validation process, the researcher selected two feature values, “love” and “excellent,” and plotted their corresponding SHAP values, as shown in Fig. 23. This allows for a clearer comparison of how each feature contributes to the model’s predictions. It further illustrates the relationship between the SHAP value of the feature “love” and its actual value across all samples (each dot representing a sample). The x-axis represents the value of the feature “love,” while the y-axis displays its corresponding SHAP value, providing insight into how variations in the occurrence of “love” influence the model’s predictions. The upward trend indicates that higher occurrences of “love” lead to more positive predictions. In addition, the plot compares the effect of the word “Excellent” on “love,” revealing that “Excellent” has a stronger impact. This suggests that the presence of “Excellent” further amplifies the positive effect of “love” on the predictions. This generally confirms that the results and findings obtained from the F-EBA model are verified and reliable, as well as qualified.

Figure 23 XAI shap-values plot for model findings validations.

Both comprehensive validation approaches (statistical and XAI validations) serve as statistical controls, tests, and evaluations of the model performances, reinforcing the validity of our results.

Fine-tuning/boosting baseline classifiers performances

As previously stated, the F-EBA employs a range of processes to generate optimized feature sets. Notably, significant accuracy improvements have been observed in the F-EBA model when utilizing these feature sets. Additionally, such optimized feature sets have been utilized to enhance/boost the accuracy of state-of-the-art classifiers so so-called baseline classifiers, without the need to modify their structures which may not always be feasible or effective in improving their performances. Many researchers have explored various boosting algorithms, from be pre-defined or custom-developed, to augment their model efficacy (Arun et al., 2018; Almouzini & Alageel, 2019; Nandy & Kumar, 2021; Safa, Bayat & Moghtader, 2022; Tong et al., 2023). Despite efforts, these methods did not contribute to improve the accuracy of baseline classifiers. On the other hand, the optimized feature set derived from this model makes a substantial contribution to boosting the performance of baseline classifiers, marking a novel advancement in the field. In this context, this study investigated the performance of the most popular state-of-the-art classifiers, including SVM, KNN, RF, NB, and LR, by utilizing both optimized feature sets and the standard bag-of-words (BOW) feature set. The overall investigation results are presented in Table 6.

Table 6 Fine-tuning/performance of baseline classifiers with optimised feature sets.

Classifiers evaluation
metrices	Shen et al. (2017)	CLPcych (2015)	
		BOW	Optimised features set	BOW	Optimised feature Set	
SMV	Accuracy	74.2	96.4	76.6	87.2	
Precision	74.9	95.0	75.3	85.9	
Recall	72.5	94.6	79.1	85.2	
F1-score	72.1	95.4	74.2	86.1	
KNN	Accuracy	73.5	82.9	73.9	81.1	
Precision	77.3	80.5	73	80.8	
Recall	74.8	81.7	74.1	79.9	
F1-score	71.4	82.1	74.5	80.1	
RF	Accuracy	69.9	83.9	59.2	83.5	
Precision	69.9	84.0	58.0	79.8	
Recall	69.9	83.9	59.2	83.5	
F1-score	69.9	83.9	58.4	72.0	
NB	Accuracy	82.2	90.6	62.6	81.2	
Precision	82.3	90.5	60.7	75.0	
Recall	82.2	92.6	62.6	81.2	
F1-score	82.2	90.5	61.1	73.8	
LR	Accuracy	69.8	78.3	63.1	63.3	
Precision	69.8	78.3	60.5	64.1	
Recall	69.8	77.3	63.1	63.1	
F1-score	69.8	78.3	60.2	65.8	

Table 6 highlights a significant enhancement in the results, with all classifiers exhibiting over a 10% improvement when utilizing optimized feature sets compared to the standard BoW in both datasets. These results signify the effectiveness of the feature engineering pipeline developed in this study for detecting depression through the generation of an optimized feature set.

Benchmarking and comparison with state-of-the-art approaches

To evaluate the efficacy of the F-EBA model, the researcher undertook a comparative analysis against state-of-the-art boosting algorithms and other ML and DL approaches on both identical datasets. This comparative analysis not only enriches our understanding of the model’s performance within the broader context of existing research but also paves the way for advancements in the field of text classification. The comparison results between the F-EBA model and various state-of-the-art approaches on the two datasets are presented in Table 7.

Table 7 Comparative analysis of studies.

Ref	Year	Algorithm	Shen et al. (2017)	Coppersmith et al. (2015)	Mental illness class	
Accuracy	F-score	Accuracy	F-score	
Shen et al. (2018)	2018	DNN-FATC	77.6	78.5	–	–	Depression	
Chiong et al. (2021)	2021	SVM, LR, DT	89.6 (Avg)	76.6 (Avg)	–	–	Depression	
de Jesús Titla-Tlatelpa et al. (2021)	2021	DT, SVM, RF	86.3 (Avg)	78.3 (Avg)	–	–	Depression	
Zogan et al. (2021)	2021	Feed-forward neural network	99.45	99.44	67.90	72.3	Depression	
Chiong et al. (2021)	2021	Gradient boosting	82.1	81.5	73.5	78.4	Depression	
Tong et al. (2023)	2022	XGboost	87.43	86.0	68.62	64.66	Depression	
Ansari et al. (2022)	2022	Ensemble hybrid learning model	--	--	65.00	65.9	Depression	
Tong et al. (2023)	2022	CBPT	88.39	68.90	70.69	66.54	Depression	
Ali et al. (2024)	2024	MentalLongformer boosting	--	92	--	73	Depression	
Our-2024		F-EBA	97.2	95.6	97.3	95.3	Depression	
Note:

The bold values highlight the performance of the proposed F-EBA model (Our-2024).

The results highlight that the F-EBA model achieved 97.3% accuracy, surpassing most state-of-the-art methods except the study conducted by Zogan et al. (2021) on a feedforward neural network achieved an accuracy of 99.45% on the Shen et al. (2017) dataset, as shown in Table 7. This higher result was achieved by testing their algorithm on a small portion (approximately 11,877 tweets) of the dataset. It’s important to note that this accuracy may decrease when the algorithm is tested on larger portions or the entire dataset. Otherwise, the results of the F-EBA model, highlight its capacity and strengths in yielding promising outcomes, particularly when confronted with larger datasets. The performance validation or finding validity of the F-EBA model against SOTA approaches was statistically tested and demonstrates that F-EBA outperforms these methods, the experimental results are shown in Table 5. Additionally, the XAI approach was employed to validate the findings, resulting in strong evidence that the model’s achieved accuracy and performance are robust and reliable.

Additionally, the difference and reasons behind the F-EBA model’s superior performance over SOTA approaches and similar algorithms, can be attributed primarily to its ability to handle complex feature sets. This capability of the F-EBA is due to utilizing advanced feature-engineering techniques, which capture and analyse complex data patterns to generate an optimized feature set, which allows the model to better understand the underlying structure of the data. Additionally, it improves the accuracy of both the F-EBA classification pipeline and state-of-the-art classifiers when compared to standard feature sets and larger datasets. Not only that, unlike Tong et al. (2023) and Laxmi Lydia, Anupama & Sharmili (2022) who utilized decision trees as a weak learner with AdaBoost, Babayomi, Olagbaju & Kadiri (2023) who combined CNN with XGBoost (C-XGBoost), Nandy & Kumar (2021) who added CatBoost into a BiLSTM layer, and Ghosal & Jain (2022) who merged FastText with XGBoost (FastText-XGBoost), this study employs two emerging algorithms, BiLSTM and BERT, as initial weak learners within the F-EBA classification pipeline. An ensemble of 10 classifiers for each is developed and boosted by utilizing F-EBA’s innovative capabilities, allowing each model to bolster its superior counterparts- subsequent models assign higher weights to misclassified features and reduce error rates. Moreover, Laxmi Lydia, Anupama & Sharmili (2022) achieved high performance in testing, they lacked full dataset cross-validation Meanwhile, Skaik & Inkpen (2020) and Tong et al. (2023) utilized 5th or 10th cross-validation, yet their results insufficient for larger datasets. To address these limitations, the performance of the F-EBA model was thoroughly evaluated across various portions of both datasets. Similarly, the use of defense mechanisms or adversarial layers allows the model to defeat similar algorithms by efficiently handling synonymous text and sarcastic posts within the dataset, resulting in improved prediction accuracy and minimizing time complexity. These are all the advantages of F-EBA over SOTA approaches. In contrast, F-EBA was tested on larger datasets using high-end computational resources (GCP with P100 GPU, 8 vCPUs, and 30 GB RAM), highlighting a disadvantage. This means F-EBA’s reliance on costly resources, which may limit its scalability and accessibility in a resource-constrained environment.

Regarding practicality, First, the F-EBA model shows substantial practicality in identifying depressive language within social media data, offering a valuable tool for mental health monitoring and early intervention efforts. It has the potential to aid mental health professionals and computational scientists in processing large volumes of user-generated content, especially in real-world clinical and research settings where extensive datasets are utilized. Second, the model’s practical adaptability allows it to be hosted online, enabling researchers to test compatible datasets efficiently. Additionally, the model’s design includes a robust defense mechanism, and its resilience against manipulated inputs intended to disrupt analysis, making it a reliable choice for practical applications. These features collectively underscore the model’s suitability for real-world applications and its readiness for integration into both research and clinical environments. However, The algorithm’s performance is currently evaluated under a controlled environment that may not fully capture the complexity of real-world conditions. In addition, the algorithm may exhibit reduced effectiveness in scenarios involving highly noisy data, muliti-lingual context, extreme class imbalance, and it may require manual parameter tuning to adapt to different domains.

Conclusion and future work

This article proposed a feature-enhanced boosting algorithm (the F-EBA model) designed for detecting depression using social media data. The model is constructed with two main pipelines: the feature-engineering pipeline, which encompasses various processes aimed at enhancing the quality of features and generating new feature sets or optimized feature sets. Secondly, a classification pipeline embeds BERT and BiLSTM models as initial weak learners and boosts them by maximizing the weight approach for misclassified samples. It also incorporates an adversarial layer or defense mechanism to withstand the accurate model against manipulated inputs designed to confuse it or against synonymous text and sarcastic posts within the datasets. The result of the feature-engineering pipeline was remarkable, producing optimized feature sets that led to substantial enhancements in performance for both the F-EBA model and state-of-the-art classifiers or baseline classifiers. This was achieved by comparative testing both the model itself and baseline classifiers, utilizing common feature sets (such as BOW, WordVec embedding, etc.), in contrast to the optimized feature sets generated by the F-EBA model. Likewise, the classification pipeline within the F-EBA model has excelled at boosting weak learners, as exemplified by BiLSTM and BERT in this study. By employing optimized feature sets and a weight maximization strategy, it has achieved notable results, reaching an impressive accuracy rate of 95%. Moreover, the integration of an adversarial layer empowered the F-EBA model to accurately handle synonymous and sarcastic text within the datasets and effectively mitigate manipulated inputs designed to confuse it. As a result of this addition, the model achieved an impressive accuracy rate of 97%, surpassing the results reported in prior studies within the field. These findings represent a notable and significant stride forward in the evolution of advanced and efficient models tailored for managing complex feature sets and synonymous and sarcastic text within extensive larger datasets. Additionally, when the model is deployed online, it exhibits the ability to effectively counteract manipulated inputs intended to cause confusion. Moreover, the validity of the F-EBA findings has been statistically tested against SOTA approaches, demonstrating its superiority, as shown in Table 5. Moreover, the XAI approach provided strong evidence that the model’s achieved accuracy and performance are robust and trustworthy. Nevertheless, for future work, additional investigation is needed to uncover potential limitations and enhance the model’s applicability across various scenarios or datasets, as well as focus on developing user-friendly applications, and exploring its integration into mental health support systems. In addition, optimising the model for faster inference or exploring its application to other types of medical imaging.

Additional Information and Declarations

Competing Interests

The authors declare that they have no competing interests.

Author Contributions

Muhammad Sadiq Rohei conceived and designed the experiments, performed the experiments, analyzed the data, performed the computation work, prepared figures and/or tables, authored or reviewed drafts of the article, and approved the final draft.

Kasturi Dewi Varathan conceived and designed the experiments, authored or reviewed drafts of the article, supervision, and approved the final draft.

Shivakumara Palaiahnakote conceived and designed the experiments, authored or reviewed drafts of the article, supervision, and approved the final draft.

Nor Badrul Anuar conceived and designed the experiments, authored or reviewed drafts of the article, supervision, and approved the final draft.

Ethics

The following information was supplied relating to ethical approvals (i.e., approving body and any reference numbers):

The Universiti Malaya Research Ethics Committee approved the study (UM.TNC2/UMREC_2148).

Data Availability

The following information was supplied regarding data availability:

The (Shen et al., 2017) dataset is available at GitHub: https://github.com/sunlightsgy/MDDL.

The CLPsych 2015 dataset is available at Johns Hopkins:

https://www.cs.jhu.edu/~mdredze/clpsych-2015-shared-task-evaluation/. To apply for access:

(1) contact Mark Dredze (mdredze@cs.jhu.edu)

(2) obtain a letter from your Institutional Review Board (IRB), or equivalent ethics board, that they have approved your proposed project and use of the data.

(3) complete the Data Use and Confidentiality Agreement.

Code is available at GitHub and Zenodo: https://github.com/sadiqllahh/FEBA-GitHub/.

Muhammad Sadiq Rohei. (2025). sadiqllahh/FEBA-GitHub: Final Release (#zenado). Zenodo. https://doi.org/10.5281/zenodo.14684890.

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
