# Peer review of "Feature-based enhanced boosting algorithm for depression detection"

_PeerJ Computer Science, doi:10.7717/peerj-cs.2981_

## Round 0.1 · original submission · Major Revisions

Dear authors,

Thank you for the submission. The reviewers’ comments are now available. It is not suggested that your article be published in its current format. We do, however, advise you to revise the paper in light of the reviewers’ concerns with respect to basic reporting, experimental design, validity of of the findings, and additional comments before resubmitting it.

Best wishes,

Reviewer 1 ·

Basic reporting

This paper proposes a feature-based enhanced boosting algorithm (F-EBA) for early detection of depression using social media data. The F-EBA algorithm aims to overcome limitations of existing boosting algorithms by optimizing feature selection and enhancing weak learners through a two-pipeline approach. Experimental results demonstrate that the F-EBA model significantly improves accuracy and performance, reaching up to 97%. However,the paper lacks sufficient detail and fails to provide a comprehensive background on the topic, with limited references to relevant literature. The claims of high accuracy and performance improvements are not adequately supported by rigorous statistical analysis or validation, raising concerns about reproducibility.

Experimental design

The experimental design of this paper relies on outdated methods and does not incorporate the advancements in large language models (LLMs). In recent years, LLMs like GPT and their variants have revolutionized the field by providing more accurate analysis capabilities. The study fails to leverage these state-of-the-art models, which could have enhanced the performance of the proposed algorithm. Secondly, the paper lacks a comprehensive discussion on related work involving large language models, missing an opportunity to contextualize the proposed approach within the broader landscape of modern machine learning techniques. This omission raises concerns about the relevance and competitiveness of the proposed methodology in light of current advancements in the field.

Validity of the findings

The study lacks a comprehensive performance comparison with state-of-the-art (SOTA) methods, not limited to those specifically tailored for depression detection. Without benchmarking against the most advanced models and techniques available, it is challenging to ascertain the true efficacy and competitiveness of the proposed algorithm. Secondly, the dataset used in the study raises concerns about its rigor and reliability. The labeling process relies merely on the presence of the word "depress" in the text, which is a simplistic and potentially misleading criterion that lacks medical or clinical validation.

Cite this review as

·

Basic reporting

The paper generally maintains a professional tone and clarity throughout the text. However, there are some grammatical errors and awkward phrasings that could hinder understanding for non-specialist readers or those not fluent in English. For instance, phrases like "unable to analyze complex sets of feature" should be corrected to "unable to analyze complex sets of features."

The article provides a robust introduction and background, situating the research within the current landscape effectively. It references a wide range of previous works, demonstrating a comprehensive understanding of the field and ongoing discussions related to depression detection using machine learning techniques.

The structure of the paper adheres to academic standards with clear, delineated sections that discuss the methodology, results, and implications in detail. Figures and tables are utilized effectively to clarify and supplement the text.

The paper is self-contained, providing all necessary information to understand the research context, methods, and findings. It successfully addresses the hypotheses posed at the beginning of the study, presenting detailed analysis and results that support the claims made.

The study appears as a coherent body of work and does not seem subdivided in a way that would inappropriately increase publication count. The research presents a new algorithm, integrates it with existing techniques, and assesses its efficacy comprehensively, making it a complete unit of publication.

Experimental design

Original Primary Research within Aims and Scope of the Journal:

The paper presents original primary research that aligns well with the aims and scope of the journal, which focuses on computational science applications. The development and testing of a new machine learning algorithm for depression detection using large datasets of social media data fit within these thematic boundaries. This topic is also highly relevant to current research interests in applying advanced computational techniques to mental health issues.

The research question is clearly defined and well-articulated. The paper addresses the need for improved algorithms for depression detection, which is both relevant and significant given the increasing reliance on digital platforms for mental health diagnostics. The paper explicitly states that it aims to fill the knowledge gap concerning the performance limitations of existing boosting algorithms when handling complex datasets and features, offering a novel solution that could potentially enhance predictive accuracy and efficiency.

The investigation appears to be rigorous and conducted to a high technical standard. The development of the Feature-Based Enhanced Boosting Algorithm (F-EBA) is detailed, with extensive testing across multiple datasets and scenarios to evaluate its efficacy and robustness. The use of a large dataset and the incorporation of adversarial training techniques suggest a technically robust approach.

On the ethical front, while the paper does not specify ethical considerations explicitly, the use of publicly available social media datasets often requires careful handling to ensure privacy and ethical compliance. It would be beneficial for the paper to discuss any ethical reviews or considerations, especially concerning data anonymization and user consent.

The methods section is detailed, providing comprehensive information on the algorithm's design, feature selection, and classification strategies. The inclusion of pseudocode and mathematical formulations adds to the reproducibility of the research. The description of the datasets, preprocessing steps, and model training parameters are generally well-detailed, which would aid replication. However, ensuring access to the exact datasets or providing more specifics on data preprocessing could further enhance reproducibility.

Validity of the findings

Impact and Novelty Not Assessed, Meaningful Replication Encouraged:

The paper does not explicitly assess its impact or novelty in the field, which aligns with the journal's policy of not basing decisions on these criteria. Instead, it focuses on the technical contribution and the functionality of the new algorithm. The replication of existing boosting techniques with significant modifications to improve performance and handling of adversarial inputs provides a clear rationale for this study. It adds value to the literature by verifying the performance of these techniques under new conditions and with a novel integration of feature sets and adversarial training, which could be seen as a form of software validation and verification.

All Underlying Data Have Been Provided; They Are Robust, Statistically Sound, Controlled:

The study mentions using a large dataset with over 46 million records and includes rigorous statistical analysis to validate the findings. The statistical methods and controls used are adequately described, and the performance of the algorithm is compared against existing methods, suggesting that the data handling and analysis are robust.

Conclusions Are Well Stated, Linked to Original Research Question and Limited to Supporting Results:

The conclusions are well-articulated and directly tied to the research questions concerning the limitations of existing boosting algorithms in analyzing complex and large datasets for depression detection. The findings are discussed in the context of the designed features of the algorithm, such as improved accuracy and robustness against adversarial attacks. The paper avoids overstating its findings and refrains from making causative claims without appropriate support. The conclusions are based strictly on the data and results presented, adhering to the principles of scientific reporting.

Additional comments

Recommendations for Improvement:

Language and Clarity: Employ a professional editing service to refine the language and correct grammatical errors.

Technical Details: Some sections might benefit from additional technical descriptions that clarify the methodology, particularly the machine learning techniques and data handling procedures.

Ethical Considerations: More explicit discussion on ethical standards, data privacy, and user consent related to the use of social media data for research purposes should be included to align with best practices in research.

Explicit Mention of Replication Value: While the paper does a good job of explaining the modifications and improvements over existing techniques, explicitly stating how this replication adds value to the existing literature could strengthen the rationale behind the study.

Further Details on Statistical Controls: Providing more explicit details about the statistical controls and tests used could help in assessing the robustness of the findings more thoroughly.

Cite this review as

Reviewer 3 ·

Basic reporting

My comments are as follows:

1. Flow of the paper is not clear.
2. The preprocessing of the dataset needs to be clear.
3. Has the evaluation metric computed through 10-fold cross validation has been done or not? Specify.
4. SOTA approaches comparison needs to be mentioned for depicting efficacy of the proposed method.
5. Proposed method needs to be supported with proper mathematical equations.
6. It is challenging to say whether any differences in model performance are significant without statistical hypothesis testing.
7. Please modify the abstract section. Write the findings only.
8. Please delete the text which is not related to the manuscript.
9. More discussion is required on recent studies.
10. What is the fundamental difference of the algorithm from the others similar algorithms?
11. More discussions are required with respect to its practicality.
12. The author(s) in this manuscript need to provide comparison of their works with the previous works on techniques used, pre-processing techniques, time complexity, type of dataset used, evaluation measures, advantages and disadvantages of their techniques.

Experimental design

As above

Validity of the findings

As above

Additional comments

As above

Cite this review as

---

## Round 0.2 · Major Revisions

Dear authors,

Thank you for the revised paper. It is still not suggested that your article be published in its current format. We do, however, advise you to revise the paper in light of the reviewer's concerns

Best wishes,

**Language Note:** The review process has identified that the English language must be improved. PeerJ can provide language editing services - please contact us at [email protected] for pricing (be sure to provide your manuscript number and title). Alternatively, you should make your own arrangements to improve the language quality and provide details in your response letter. – PeerJ Staff

Reviewer 4 ·

Basic reporting

- Clear and unambiguous, professional English used throughout: The manuscript is not written in professional English. In addition, there are many typos as in the abstract section (boos t). Some sections contain overly complex sentences that can be restructured for better clarity and readability. You should check them. For example:
"The optimized feature sets derived from the F-EBA model substantially contribute to boosting the performance of baseline classifiers, marking a novel advancement in the field." This could be revised for conciseness: "The optimized feature sets from F-EBA improve baseline classifier performance, demonstrating a significant advancement."
A language revision to enhance fluency and clarity is recommended. The authors may benefit from professional proofreading.
- Literature references, sufficient field background/context provided: The study includes an extensive literature review on boosting algorithms, feature selection techniques, and depression detection models. However, recent state-of-the-art methods (published after 2022) should be compared in more detail. A comparison table summarizing previous studies and how the proposed method differs is recommended.
- Professional article structure, figures, tables. Raw data shared: The manuscript adheres to standard PeerJ structure. Figures and tables are relevant but lack detailed captions explaining their significance. Dataset details should be provided in a dedicated "Dataset" subsection. Sample dataset images should be included to illustrate the types of features extracted.
- Self-contained with relevant results to hypotheses: The study is self-contained and presents all relevant findings. However, the hypothesis is not explicitly stated, and it would be beneficial for the authors to clearly define their research question in the introduction. Although authors emphasize the limitations of the field, the introduction section does not adequately provide the motivation and aim of the study. The study's contribution to existing research should be clarified in the Introduction. In abstract section, you should emphasize the used features, models and their contributions.

Experimental design

- Original primary research within Aims and Scope of the journal: The study aligns with PeerJ’s Aims & Scope by presenting an advanced computational approach to depression detection. The research question is well-defined and relevant. The research question is clear and well-defined. The novelty is weak. The methods used are well known and does not contribute to computer science.
- Research question well defined, relevant & meaningful. It is stated how research fills an identified knowledge gap: The study effectively identifies gaps in existing boosting algorithms but should explicitly state how F-EBA addresses them. The comparison with state-of-the-art models needs improvement, including models published after 2023.
- Rigorous investigation performed to a high technical & ethical standard: The feature engineering process is well-explained, but hyperparameter settings are missing. The computational complexity of F-EBA should be discussed. The evaluation environment should be described, including hardware specifications and runtime efficiency. The complexity of the model should be calculated.
- Methods described with sufficient detail & information to replicate: The methodology is mostly reproducible, but the following improvements are needed: Detailed hyperparameter settings for BiLSTM, BERT, and boosting framework. Explanation of how parameters were optimized for best performance. Justification for choosing BiLSTM and BERT over other deep learning models. Computational efficiency analysis including training and inference time.

Validity of the findings

- Impact and novelty not assessed. Meaningful replication encouraged where rationale & benefit to literature is clearly stated: The study presents strong experimental results but lacks an explicit discussion on reproducibility. The authors should explain how F-EBA can be generalized to other datasets.
- All underlying data have been provided; they are robust, statistically sound, & controlled: Statistical significance tests should be included to support the reported improvements. Confidence intervals, p-values, or effect size calculations should be provided.
- Conclusions are well stated, linked to original research question & limited to supporting results: The conclusions accurately reflect the findings, but some claims need to be softened. For example: "F-EBA establishes itself as the most powerful model for depression detection." This should be reworded to "F-EBA demonstrates strong potential for depression detection, outperforming baseline models."

Additional comments

This review critically evaluates the article titled "Feature-Based Enhanced Boosting Algorithm for Depression Detection." The study presents a Feature-Based Enhanced Boosting Algorithm (F-EBA) to improve the accuracy and interpretability of depression detection models based on social media data. The proposed model integrates feature engineering and ensemble learning techniques, leveraging BERT, BiLSTM, and an adversarial layer to enhance classification performance. The study claims significant accuracy improvements, achieving up to 97% accuracy. However, several key aspects require further clarification and improvements. This review assesses the novelty, technical soundness, contribution to the field, and overall relevance of the work. The suggested revisions outlined below are expected to enhance the clarity, rigor, and impact of the article.
1. Give some sample datasets to better understand. Increase explainability of features. The article should provide a detailed description of the dataset properties in a different subsection titled “dataset”.
2. How did you enhance the boosting algorithm? What is your contribution to boosting algorithm?
3. The “Discussions" section should be added in a more highlighting, argumentative way. The evaluation results should be described in more details including the discussion about the algorithm complexity. The discussion section could benefit from an exploration of potential limitations, such as computational demands or limitations in real-time processing.
4. The limitations of the algorithm and the evaluation environment should be discussed in the paper. What are the capabilities, benefits and limitations of study?
5. Suggestions for future research, like optimizing the model for faster inference or exploring its application to other types of medical imaging, would add depth to the study.
6. Explain why your model is obtains low accuracy values. Compare them state of the art models published after 2024.
7. The complexity of the proposed model and the model parameter uncertainty are not enough mentioned.
8. The authors should consider discussing the computational efficiency of the model, including training time and inference time.
9. The article lacks comparisons with state-of-the-art methods, specifically those developed after 2022. The authors should compare the results of their method with those of previous studies. As mentioned in the literature, there are several methods with very high accuracy, even better than the proposed method. Author(s) can do compare table (A new table can add about previous studies to result section.).
10. How did you set the parameters of proposed method for better performance?
11. As a performance metrics, confusion metrics (f1 score, accuracy rate, recall, precision etc…) can be given and accuracy rate should be considered as a first metric.
12. The authors need to show clearly what their contribution is to the body of knowledge.
13. The performance of the proposed method should be better analysed, commented, and visualized in the experimental section.
14. Improve writing clarity by simplifying complex sentences.
15. Justify improvements with statistical significance tests.
16. Discuss limitations more explicitly, especially regarding emotion misclassification.

Cite this review as

---

## Round 0.3 · Minor Revisions

Dear Authors,

Thank you for the revised paper. We encourage you to address the minor concerns and criticisms of Reviewer 5 and resubmit your paper once you have updated it accordingly.

Best wishes,

Reviewer 4 ·

Basic reporting

The authors have addressed my concerns about the article.

Experimental design

The authors have addressed my concerns about the article.

Validity of the findings

The authors have addressed my concerns about the article.

Additional comments

The authors have addressed my concerns about the article.

Cite this review as

Reviewer 5 ·

Basic reporting

In this study, the authors develop a new machine learning algorithm, Feature-Based Enhanced Boosting Algorithm, to detect depression more successfully on social media data. Two large, labeled datasets consisting of millions of tweets were used to test the model. First, data cleaning was applied. Then, text normalization was performed. Emoji and visual content were converted into undeleted text. Then, the tokenization and lemmatization steps were used. Swear words were preserved, and the extreme class was reduced to remove the imbalance. Vectorization was applied with Bert and Word2vec methods. Word2Vec does not consider direct contextual relations, so the contribution of each word is considered equal. Different importance levels are assigned to words with the attention mechanism to overcome this deficiency. The features generated by the embeddings are evaluated in a decision tree structure, and RFE is applied. The F-EBA model was compared with SVM, RF, KNN, NB, LR, GRU, and LSTM models. F-EBA gave the best result. The study is original due to the new model it proposes and the successful results it obtained. The following changes should be made:

-A separate section on motivation and contribution should be added to the introduction. The motivation and contribution of the study should be explained under this subsection.

- The following English paragraph in the introduction should be rewritten more simply and clearly. “Feature Optimization using the F-EBA feature engineering pipelines, which leverage Word2Vec and BERT models, self-attention mechanism, and recursive feature elimination (RFE). This process extracts and weighs features, eliminating those with low weights, and ultimately generating an optimized feature set.”

Experimental design

.

Validity of the findings

.

Cite this review as

---

## Round 0.4 · accepted · Accept

Dear Authors,

The last reviewer who suggested minor revision did not respond to the invitation for reviewing the revised paper. It seems that the paper has been improved according to the reviewer's comments and is ready for publication.

Best wishes,